# Dibenzocyclooctadiene Lignans from *Schisandra chinensis* with Anti-Inflammatory Effects

**DOI:** 10.3390/ijms25063465

**Published:** 2024-03-19

**Authors:** Michal Rybnikář, Milan Malaník, Karel Šmejkal, Emil Švajdlenka, Polina Shpet, Pavel Babica, Stefano Dall’Acqua, Ondřej Smištík, Ondřej Jurček, Jakub Treml

**Affiliations:** 1Department of Natural Drugs, Faculty of Pharmacy, Masaryk University, 61200 Brno, Czech Republic; 507131@mail.muni.cz (M.R.); malanikm@pharm.muni.cz (M.M.); smejkalk@pharm.muni.cz (K.Š.); svajdlenkae@pharm.muni.cz (E.Š.); 461298@mail.muni.cz (P.S.);; 2Recetox, Faculty of Science, Masaryk University, 62500 Brno, Czech Republic; pavel.babica@recetox.muni.cz; 3Department of Pharmaceutical and Pharmacological Sciences, University of Padova, 35131 Padua, Italy; stefano.dallacqua@unipd.it; 4Department of Molecular Pharmacy, Faculty of Pharmacy, Masaryk University, 61200 Brno, Czech Republic; 474476@mail.muni.cz

**Keywords:** anti-inflammatory, antioxidant, dibenzocyclooctadiene, gap junction, lignan, *Schisandra chinensis*

## Abstract

*Schisandra chinensis* (Schisandraceae) is a medicinal plant widely used in traditional Chinese medicine. Under the name Wu Wei Zi, it is used to treat many diseases, especially as a stimulant, adaptogen, and hepatoprotective. Dibenzocyclooctadiene lignans are the main compounds responsible for the effect of *S. chinensis*. As a part of ongoing studies to identify and evaluate anti-inflammatory natural compounds, we isolated a series of dibenzocyclooctadiene lignans and evaluated their biological activity. Furthermore, we isolated new sesquiterpene 7,7-dimethyl-11-methylidenespiro[5.5]undec-2-ene-3-carboxylic acid. Selected dibenzocyclooctadiene lignans were tested to assess their anti-inflammatory potential in LPS-stimulated monocytes by monitoring their anti-NF-κB activity, antioxidant activity in CAA assay, and their effect on gap junction intercellular communication in WB-*ras* cells. Some *S. chinensis* lignans showed antioxidant activity in CAA mode and affected the gap junction intercellular communication. The anti-inflammatory activity was proven for (−)-gomisin N, (+)-γ-schisandrin, rubrisandrin A, and (−)-gomisin J.

## 1. Introduction

*Schisandra chinensis* (Turcz.) Baill. (Schisandraceae) belongs to 50 essential herbs of traditional Chinese medicine (TCM) [1]. It is grown on large plantations in China, Korea, Japan, and Russia (the Primorsky, Khabarovsky regions, the Kuril and Sachalin islands) [2]. In China, *S. chinensis* is called Wu Wei Zi, which means the fruits of five tastes: sweet, salty, bitter, sour, and astringent [3]. *S. chinensis* is a deciduous, shrub-like, dextrorotary creeper and it grows to a height of 10–15 m [4]. The fruit forms an elongated cluster of red berries about 5–7.5 mm in size and contains 1–2 kidney-shaped seeds [5]. The best quality and largest fruits come from the northeastern provinces of China, named northern Schisandra—Bei Wu Wei Zi [6].

Dibenzocyclooctadiene lignans are the most important active substances from *S. chinensis* with biological and pharmacological activity. Lignans consist of two phenylpropane units linked through the propane chain central carbons (β) [7]. Phytochemical works have so far described several tens of schisandra lignans, which differ from each other both in the conformation of the polycyclic system and in the presence of different substituents, both on the biphenyl and on the cyclooctadiene ring. Dibenzocyclooctadiene lignans of *S. chinensis* are found in various concentrations throughout the plant, mainly in seeds and fruits [8].

*S. chinensis* is used both for medicinal purposes and as a food ingredient due to its specific taste, and besides the cyclooctadiene lignans, it contains organic acids (citric, malic, fumaric, and tartaric acid), microelements (Cu, Mn, Ni, Zn, Mo), vitamins (A, B, C, D, E), sugars, triterpenes, flavonoids, and essential oils [2,4]. *S. chinensis* is used as an additive to increase the taste and nutritional value of food, and also as a preservative [9]. One hundred (100) g of dried fruits contains Fe, Mn, Cu, K, and Mg in amounts that cover 96%, 320%, 48%, 54%, and 33% of the recommended daily intake, respectively [10].

The activity of dibenzocyclooctadiene lignans has been investigated in many studies that have confirmed adaptogenic effects, central nervous system stimulation, hepatoprotective effects, anti-inflammatory, and potential anticancer, as reviewed in [2,4]. *S. chinensis* is in TCM used for the treatment of diarrhea, sweating, shortness of breath, palpitations, anxiety, liver injury, pneumonia, heart diseases, hypertension, chronic fatigue syndrome, deterioration of memory, depression, stress, and it is an essential adaptogen [2,4]. The main active substances of *S. chinensis* that have antioxidant and anti-inflammatory activity are lignans [11].

Oxidative stress is a process where reactive oxygen and nitrogen species (ROS and RNS) are produced initially for protective and regulatory purposes, but when they become uncontrollable, they cause damage to the organism [12,13]. ROS and RNS may contribute to inflammation, Parkinson’s, and Alzheimer’s disease, cancer, atherosclerosis, damage DNA, diabetes mellitus, hepatitis, and other chronic and degenerative diseases [14,15]. The antioxidant activity of dibenzocyclooctadiene lignans is associated with the level of glutathione and the activity of glutathione peroxidase. It is linked with the inhibition of lipoperoxidation and the formation of cellular peroxides [16,17,18].

Inflammation is the basic process by which the body reacts to irritation, infection, and damage [19]. Activation of nuclear factor-kappa B (NF-κB) plays a crucial role in inflammation through the ability to induce transcription of pro-inflammatory genes, including those controlling the production of cytokines and chemokines. The NF-κB pathway can be activated by pathogens or stress factors [20]. Dibenzocyclooctadiene lignans of *S. chinensis* reduced the phosphorylation of inhibitor of NF-κB kinase subunit alpha and beta, NF-κB inhibitor alpha (IκBα), protein kinase B (Akt), TANK-binding kinase 1, extracellular signal-regulated kinase (ERK), p38 mitogen-activated protein kinase (MAPK), c-Jun N-terminal kinase (JNK), NF-κB subunit p65, activator protein 1 of the transcription factor c-Jun, and interferon regulatory factor 3 (IRF3) in RAW264.7 macrophages activated with LPS and thus suppressed the NF-κB pathway [21]. Lignans can affect inflammation also by influencing other mechanisms [9,22].

Gap junction intercellular communication (GJIC) allows communication between cells and the exchange of small molecules through connexin-based gap junction channels [23]. Gap junction-coupled cells can be found in various organs in the body, where GJIC helps to maintain tissue homeostasis and protects the cells from harmful substances and conditions such as oxidative stress or inflammation [24]. Disruption and dysregulation of communication by inflammatory mediators or endotoxins (lipopolysaccharides-LPS) can cause inflammatory substances to spread to other cells and organs in the body and can lead to systemic inflammation [25]. Furthermore, disruption of GJIC can be a factor promoting cancerogenesis. Substances capable of increasing GJIC are coming to the fore. An important group consists of plant phenols, which include lignans, flavonoids, phenolic acids, and stilbenes. It was shown that plant phenolic compounds can modulate drug-metabolizing enzymes, inhibit oxidative stress, or enhance GJIC suppressed in tumor-transformed cells [26].

Based on the previously described findings, we isolated and identified eight dibenzocyclooctadiene lignans and two aryltetralones. Furthermore, the chromatographic separation of the lipophilic portion of *S. chinensis* fruit extract led to the isolation of new bicyclic sesquiterpene, namely 7,7-dimethyl-11-methylidenespiro[5.5]undec-2-ene-3-carboxylic acid (**1**). The selected isolated compounds, together with some previously isolated schisandra lignans, were assayed to verify the ability of lignans to influence GJIC in WB-*ras* cells and to evaluate the antioxidant and anti-inflammatory activity of dibenzocyclooctadiene lignans.

## 2. Result and Discussion

### 2.1. Isolation of Compounds

An investigation of compounds of the fruit of *S. chinensis* led to the isolation and structural determination of eight dibenzocyclooctadiene lignans and two aryltetralones. These compounds were isolated by extensive chromatographic separation both in normal and reversed phase and identified based on a comparison of their ^1^H and ^13^C NMR spectra, MS, CD, and IR spectra with corresponding literature as (−)-gomisin G (**2**) [27], (−)-gomisin B (**3**) [27], (−)-gomisin C (**4**) [27], (−)-angeloylgomisin P (**5**) [28], epigomisin O [29], arisantetralone C [30], arisantetralone A [30], and (−)-schisantherin E [31]. (+)-Deoxyschisandrin (**6**) and (−)-wuweizisu C were identified by comparison with previously obtained lignan standards [32] (Figure 1). Arisantetralones A and C were isolated from *S. chinensis* for the first time.

New sesquiterpene **1** was isolated and identified from fraction SC 183-205-X-XI-D obtained by the chromatographic fractionation of chloroform portion of *S. chinensis* fruit extract. The compound **1** was isolated as an amorphous solid in the amount of 2 mg. Analysis of the accurate mass spectra of compound **1** in the negative mode (Appendix A) showed the molecular ion [M–H]^−^ observed at *m*/*z* 233.15404, corresponding to C_15_H_21_O_2_^−^ and allowed us to establish the formula to be C_15_H_22_O_2_ (calcd 234.16131).

Compound **1** showed the presence of two singlets in the aliphatic region, each integrating for three protons (δ_H_ 0.91 and 0.87 ppm) in ^1^H-NMR spectrum, suggesting the presence of two methyl groups connected to quaternary carbon, multiplets in the region of δ_H_ 1.40–2.34 ppm supporting the presence of aliphatic moiety, a broad singlet at δ_H_ 6.96 ppm integrating for one proton, and two doublets at δ_H_ 4.84 and 4.41 ppm (*J* = 1.8 Hz) suggesting the presence of two geminal proton sp^2^ signals. The HSQC DEPT spectrum allowed us to assign the H and C of non-quaternary position, and results indicated that there are two quaternary-carbon connected methyl groups, five aliphatic methylenes, one sp^2^ methylene and one sp^2^ CH (see Table 1). The HMBC spectrum allowed us to establish the structure of compound **1**. Diagnostic HMBC correlations were observed from methyl groups C-12 and C-13 with carbon resonances at δ_C_ 36.7 (C-9), δ_C_ 25.4 (C-8), and δ_C_ 44.6 ppm (C-4). Other diagnostic correlations were observed from the H-7 (δ_H_ 2.29 ppm) with C-8 and C-9, and from H-6 (δ_H_ 1.79–1.95 ppm) with C-8, C-7 (δ_C_ 29.6 ppm), C-5 (δ_C_ 148.0 ppm), and C-15 (δ_C_ 110 ppm) suggesting the presence of exocyclic methylene, and C-4 supporting the existence of a cyclohexene ring. The position C-4 is quaternary and is visible from the signals H-3 (δ_H_ 2.15–2.26 ppm), H-10 (δ_H_ 1.80–1.96 ppm), and H-11 (δ_H_ 2.34–2.36 ppm), suggesting that the structure is a spiro compound. Further diagnostic signals indicated the presence of a second cyclohexene ring as supported by the HMBC correlations observed from H-10 with C-4, C-9, and C-5, confirming the spiro linkage, and from the correlation of the H-10 with C-1 (δ_C_ 138.4 ppm) and C-3 (δ_C_ 31.6 ppm). Further diagnostic correlations were observed on the HMBC spectrum from H-11 with C-4, C-2 (δ_C_ 140.6 ppm), and C-14 (δ_C_ 175.0 ppm), suggesting the presence of an unsaturated alpha acid moiety (Appendix A). Thus, the structure of compound **1** was established as 7,7-dimethyl-11-methylidenespiro[5.5]undec-2-ene-3-carboxylic acid (Figure 2).

After evaluating the yields of isolated substances, a combination of isolated compounds and lignans which we isolated from *S. chinensis* previously [32] was selected for further tests of their antioxidant and anti-inflammatory properties, and GJIC: (−)-gomisin G (**2**), (−)-gomisin B (**3**), (−)-gomisin C (**4**), (−)-angeloylgomisin P (**5**), (+)-deoxyschisandrin (**6**), (+)-schisandrin (**7**), (−)-gomisin N (**8**), (−)-tigloylgomisin P (**9**), (−)-tigloyldeangeloylgomisin F (**10**), (+)-gomisin A (**11**), (+)-γ-schisandrin (**12**), rubrisandrin A (**13**) (an inseparable mixture of regioisomers **13a** and **13b** in a 2:1 ratio), and (−)-gomisin J (**14**) (Figure 1).

### 2.2. Antioxidant Activity of Lignans

The antioxidant activity was tested using the cellular antioxidant activity (CAA) assay. CAA was developed to evaluate the activity of natural substances, foods, and dietary supplements to better represent the conditions found in living organisms [33]. The antioxidant activity of compounds **2**–**14** was tested and compared with the standard antioxidant used—flavonoid quercetin (Figure 3)—which demonstrated the highest CAA value [34]. The compounds were tested at a non-toxic concentration of 10 µM (viability above 80%), as shown by antiproliferative WST-1 assay (Appendix A).

*S. chinensis* and its extracts could inhibit oxidative damage of DNA and cells induced by hydroxyl radicals via its antioxidant properties [35]. Lignans of *S. chinensis* in rat liver microsomes induced antioxidative enzymes that inhibited peroxidation of lipids induced by iron/cysteine, measured by means of a decrease in malondialdehyde [17]. Previously published works confirmed the antioxidant protective effect of lignans on liver cells [4,5].

No tested compound showed significant antioxidant activity in CAA assay, except for (−)-gomisin J (**14**) and (−)-tigloylgomisin P (**9**), with CAA values reaching approximately a half and a quarter, respectively, of the activity of quercetin (*p* < 0.001 and *p* < 0.05, respectively.). The activity of compound **14** is probably due to two free hydroxyl groups that can act as proton donors. A previous study showed that the increased antioxidant activity of modified-ultrasonic *S. chinensis* extract is possibly associated with the presence of (−)-gomisin J (**14**) and (−)-tigloylgomisin P (**9**) [36], which is supported by our results. Furthermore, previously published studies have shown that (−)-gomisin J (**14**) suppressed antioxidative stress and lipid peroxidation [37] and protected against hydrogen peroxide-induced cell death [38]. The negative CAA values are a sign of a pro-oxidant effect of such compounds. The most prominent in this regard is (+)-schisandrin (**7**).

The ability to donate a proton predetermines the antioxidant property of a molecule [39]. Compound **9** showed some degree of antioxidant activity, although it lacks free hydroxyl groups that can act as proton donors. Previously published literature showed that the antioxidant activity of lignans depends also on their stereoconfiguration. For lignans with the same molecular structures but different stereoconfiguration, the *S* arrangement of the benzene ring has a stronger quenching effect of free oxygen radicals than the *R* arrangement [40]. Furthermore, the presence of exocyclic methylene group and the benzoyloxy group may slightly increase the activity [16]. Both groups were found only for *S*-biphenyl configuration of dibenzocyclooctadiene lignans [16]. The antioxidant activity is also affected by the position and number of tigloyl, angeloyl, hydroxyl, and benzoyl groups on the cyclooctadiene ring [16]. It was previously reported that (+)-gomisin A (**11**) [41] and (−)-gomisin C (**4**) [42] showed good antioxidant effects, but it was not confirmed in our experiment. Also, other previously published studies described (+)-deoxyschisandrin (**6**) (syn. schisandrin A) [43] and γ-schisandrin (**12**) (syn. schisandrin B) [44] as good antioxidants, but we have not confirmed these results. The differences between our results and the presented studies might be explained by different methodologies used. Oxidative stress can be induced with high glucose levels [41], H_2_O_2_ [43], or *tert*-butyl hydroperoxide [44]. You et al. did not generate stress but measured the ability of compound **4** to decrease the basal level of stress in adipocytes [42].

### 2.3. Anti-Inflammatory Activity

The anti-inflammatory activity of the test lignans **2**–**14** was evaluated in THP1-Blue™ NF-κB cells and compared to the prednisolone used as a positive control. The concentration of the test compounds was selected based on our previous results, and any test compounds showed cytotoxicity at tested concentration. (−)-Gomisin N (**8**) and (+)-γ-schisandrin (**12**) showed a statistically significant anti-inflammatory effects (*p* < 0.0001) in a concentration of 10 µM and their activity was even higher compared to prednisolone (*p* < 0.001) used in standard concentration of 2 µM. (−)-Angeloylgomisin P (**5**) showed significant activity (*p* < 0.001). Moreover, (+)-deoxyschisandrin (**6**), (−)-tigloylgomisin P (**9**), rubrisandrin A (**13a** and **13b**) and (−)-gomisin J (**14**) also showed statistically significant effects (*p* < 0.01) (Figure 4). Lignans with *S*-biphenyl configuration and methylenedioxy groups had stronger anti-inflammatory activity. Also, the methoxy group on the cyclooctadiene increased effectiveness. Our findings agree with the previous results, showing that lignans with the *S*-configuration (**5**, **8**, **9**, **13**, and **14**) show greater activity than those with *R*-configuration (**7**, **11**) [45]. We confirmed this finding with (−)-gomisin N (**8**), which possesses *S*-configuration.

The anti-inflammatory properties of dibenzocyclooctadiene lignans have been investigated in several studies, however, many previous reports do not mention the configuration of biphenyl and optical rotation. Therefore, for these discussed reports we do not state (+) and (−) isomers and numbers of structures. Previous results proved that the anti-inflammatory effect of lignans is related to the inhibition of nitric oxide (NO) formation, inhibition of expression of NO-synthase (NOS) [46], and the decrease in production of prostaglandin E2 (PGE2), which may be related to down-regulation of iNOS and cyclooxygenase 2 (COX-2) expression [21]. γ-Schisandrin suppressed the activation of NF-κB induced by LPS and increased the expression of transcription factor Nrf2 [47]. γ-Schisandrin also reduced the expression of tumor necrosis factor (TNF-α) and interleukin 1β (IL-1β) in myocardial tissues [48] and suppressed the production of IL-6, IL-10, and IL-12 induced by LPS in dendritic cells and decreased glutathione levels in human monocytic THP-1 cells [49]. (+)-γ-Schisandrin (**12**) inhibited the phosphorylation of IκBα, NF-κB p65 subunit, MAPK, and JNK [50].

γ-Schisandrin and (+)-deoxyschisandrin (**6**) inhibited inflammation by suppressing IL-1β secretion and pyroptosis by inhibiting the NLR family pyrin domain containing 3 inflammasome activation in *Propionibacterium acnes*-infected THP-1 cells, with γ-schisandrin acting stronger than **6** [51], which matched with our results. In a previous study, γ-schisandrin significantly inhibited LPS-induced NF-κB activation [52], which fully agreed with our results and demonstrated its anti-inflammatory activity.

Previously, (+)-deoxyschisandrin (**6**) reduced secretion of pro-inflammatory cytokines (TNF-α, Il-1β) induced by LPS. It also inhibited NF-κB, MAPK, and phosphatidylinositol-3 kinase)/Akt pathways [53]. Furthermore, (+)-deoxyschisandrin (**6**) inhibited IL-1β-induced inflammation and cartilage degradation by inhibiting the NF-κB signalling pathway in rat chondrocytes and suppressed the production of NO and prostaglandin E2 [54].

(−)-Gomisin N (**8**) and (−)-gomisin J (**14**) decreased mRNA expression and reduced the secretion of pro-inflammatory cytokines. The inhibitory activity was related to the inhibition of MAPK, ERK 1/2, and JNK phosphorylation [45]. (−)-Gomisin N (**8**) and (−)-gomisin J (**14**) reduced the production of NO induced by LPS in RAW 264.7 cells [46] and **8** also reduced levels of the iNOS protein [55]. It was shown that lignans can inhibit NO induction in IL-1*β*-stimulated hepatocytes [55] in the same order of magnitude as was achieved in our anti-NF-κB assay (**8** > **12** > **6**). The exact mechanism by which (−)-gomisin N (**8**) inhibits NF-κB is not known. It was proposed that it inhibits phosphorylation of IκB kinase (IKK) and thus suppresses activation of NF-κB. Moreover, (−)-gomisin N (**8**) might inhibit mitogen-activated protein kinase kinases (MAP2Ks) or suppress phosphorylation of Akt/protein kinase B, in both cases resulting in inhibited NF-κB [55]. (−)-Gomisin N (**8**) possesses some 15-lipooxygenase, COX-1, and COX-2 inhibitory activities [56] and showed antioxidant, anti-inflammatory, and hepatoprotective activity in vivo and in vitro [57]. Furthermore, our results agree with the previously published effects on a decrease in iNOS and COX-2 levels (**8** > **14** > **2** > **11**) [58]. (−)-Gomisin J (**14**) reduced the levels of cleaved caspase-3, Bax protein, COX-2, NO, and NF-κB in ischemia/reperfusion injury rat brain tissues [37].

### 2.4. Gap Junction Intercellular Communication (GJIC)

We tested the ability of lignans **2**–**14** to increase GJIC in the WB-*ras* cell line. A starting concentration of 10 µM was used. We observed only moderate to low effects of the test lignans (Figure 5) in comparison with the positive control sodium butyrate, which is a differentiation-promoting compound capable of inducing GJIC. GJIC levels were reduced by (−)-gomisin G (**2**), probably because of its growth-inhibiting or cytotoxic activity as evidenced by incomplete confluency, which in turn prevented proper GJIC quantification. The highest activity showed (+)-schisandrin (**7**) at a concentration of 10 µM, and activity was statistically significant (*p* < 0.001) (Figure 6). Based on the evaluation of the obtained results at a concentration of 10 µM, we selected (+)-schisandrin (**7**), (−)-gomisin N (**8**), and (−)-tigloylgomisin P (**9**) at concentrations of 5 µM and 20 µM for additional evaluation. For these compounds, the change in concentration did not significantly affect the observed activity (data shown only for **7**). In fact, the activity of compound **7** appeared to peak at 10 µM, while its effects at 20 µM were relatively less potent. This could be attributed to a decrease in cell viability with an increasing concentration. Such a phenomenon was previously observed, for example, with a GJIC-inducing natural bioactive phenolic compound, caffeic acid phenyl ester [59].

This is the first report on GJIC for dibenzocyclooctadiene lignans. Only one study reported that DDB (4,4′-dimethoxy-5,6,5′,6′-dimethylendioxy-biphenyl-2,2′-dicarboxylate), a synthetic analogue of (−)-gomisin C (**4**), attenuated 12-*O*-tetradecanoylphorbol-13-acetate (TPA)-induced down-regulation of GJIC. The result suggested that DDB prevented oncogenic transformation of WB-F344 cells [60].

Regarding touching other lignans, a previous study showed that secoisolariciresinol from flaxseed can interfere with the cellular activity of malignant tumors, which affects GJIC, and regulates related signaling pathways [61].

## 3. Material and Methods

### 3.1. Plant Material

The fruits of *S. chinensis* (produced in northeastern Dongbei region (China) were purchased from Salvia Paradise Ltd. (Prague, Czech Republic) in 2021. Chromatographic fraction SC 183–205 from chloroform fraction was obtained previously [62]. Lignan standards (−)-gomisin G (**2**), (+)-deoxyschisandrin (**6**), (−)-wuweizisu C, (+)-schisandrin (**7**), (−)-gomisin N (**8**), (−)-tigloylgomisin P (**9**), (−)-tigloyldeangeloylgomisin F (**10**), (+)-gomisin A (**11**), rubrisandrin A (**13**), and (−)-gomisin J (**14**) were obtained as described previously [32].

### 3.2. General Analytical Apparatus and Conditions

UV spectra were obtained using an Agilent 1100 chromatographic system with an Agilent 1100 Series Diode Array Detector (Agilent Technologies, Waldbronn, Germany). Circular dichroism spectra were recorded on a JASCO J-815 CD spectrometer (Jasco, Tokyo, Japan). IR spectra (ATR technique) were measured with a Bruker Alpha FT-IR Spectrometer (Bruker Daltonik, Bremen, Germany). 1D and 2D NMR spectra were obtained on a JEOL ECZR 400 MHz NMR spectrometer (JEOL, Tokyo, Japan) with TMS as the internal standard for all compounds with except of **1** (Bruker Avance 3 400 MHz for ^1^H and 100 MHz for ^13^C, Bruker, Germany). Compounds were measured in CDCl_3_ (Sigma-Aldrich, Steinheim, Germany) (**2**, **3**, epigomisin O, arisantetralone C), CD_3_OD (Sigma-Aldrich) (**1**, **4**, **5**) and DMSO-*d*_6_ (Eurisotop, Paris, France) (arisantetralone A, (−)-schisantherin E). Compound **1** was analyzed by the Agilent 1260 Infinity LC system connected to the mass spectrometer Orbitrap Elite with Velos Pro (Thermo Fisher Scientific, Bremen, Germany). As a column was used the Ascentis^®^ Express 90 Å RP-Amide HPLC column (100 × 2.1 mm, 2 µm particles). The injection volume was 5 µL and the flowrate was 0.2 mL/min. Mobile phase A was water with 0.2% formic acid. Mobile phase B was acetonitrile. The LC gradient was used as follows: 0 min B = 10%, 36 min B = 100%, 50 min B = 100%, 50.1 min B = 10%, 60 min B = 10%, STOP. Column was thermostated at 30 °C. The mass spectrometer used FTMS (Orbitrap, Thermo Fisher Scientific, Bremen, Germany) negative ion mode with two scan events, where the first scan detected MS1 spectra with a resolution 60,000 and in a mass range 120–1200 *m*/*z*, and the second scan used data-dependent analysis with a minimal signal threshold of 50000, activation type HCD, normalized collision energy 55 and activation time 0.1 s. LRMS data of further compounds were recorded using a LC-MS/MS system consisting of AB SCIEX Triple Quad 3500 System (AB SCIEX, Framingham, MA, USA) coupled with a UPLC Ultimate 3000 chromatographic system (Thermo Fischer Scientific, Waltham, MA, USA), in both the positive and negative modes. Analytical HPLC measurements were carried out with an Agilent 1100 chromatographic system (Agilent Technologies, Darmstadt, Germany) and Dionex UltiMate 3000 HPLC System (Thermo Fischer Scientific, USA) with Varian 380-LC ELSD (Varian, Church Stretton, UK). Semi-preparative RP-HPLC was performed using a Dionex UltiMate 3000 HPLC System with fraction collector (Thermo Fischer Scientific, Germering, Germany) and a YL 9100 HPLC System (Young Lin, Anyang, The Republic of Korea) with a FOXY R2 fraction collector (Teledyne Isco, Lincoln, NE, USA).

Compounds were separated by column chromatography using silica gel with a particle size of 40–63 μm (Merck, Steinheim, Germany). Further semi-preparative HPLC separations were performed with an Ascentis RP-Amide, 25 cm × 10 mm, particle size 5 μm, semi-preparative HPLC column (Sigma-Aldrich) or an Ascentis C18, 25 cm × 10 mm, particle size 5 µm, semi-preparative HPLC column (Sigma-Aldrich). Silica gel 60 F_254_ (20 × 20 cm, 200 μm) TLC plates (Merck) and an Ascentis Express RP-Amide, 10 cm × 2.1 mm, particle size 2.7 μm, analytical HPLC column (Sigma-Aldrich) were used for analytical purposes. Gradient grade MeCN and MeOH for HPLC were purchased from Sigma-Aldrich or VWR International (Briare, France), and other analytical grade solvents from Lach-Ner (Neratovice, Czech Republic).

### 3.3. Extraction and Isolation

*S. chinensis* dried fruits (1 kg) were crushed and extracted by maceration in MeOH (4 × 2 L, 4 × 4 days) facilitated by ultrasonification (3 × 1 h, temperature 40 °C, frequency 35 kHz; Sonorex Digitec, Bandelin electronic GmbH & Co. KG, Berlin, Germany). The extract was evaporated until dry under a vacuum and dissolved in MeOH (90%, *v*/*v*) with water (1 L). The extract was shaken with *n*-hexane to remove lipophilic interfering substances in a ratio of 1:3 (hexane: 90% methanol, *v*/*v*), thus obtaining *n*-hexane and methanol fraction, which was further processed. Methanol was evaporated from the methanol fraction, and then the extract was dissolved in water, chloroform was added in a ratio of 1:2 (water:chloroform, *v*/*v*, total volume 1 L) and shaken (5× repeated). The chloroform fraction was dried under vacuum, and 19.3 g of chloroform extract was consequently separated by silica gel column chromatography (glass column i.d. 7.4 cm × 94 cm) using a stepped gradient of a mobile phase composed of methanol:*n*-hexane:chloroform, starting composition 0.5:5:94.5 (*v*/*v*/*v*) to ending composition 30:5:65 (*v*/*v*/*v*). Fractions of 150 mL were collected. In total, 283 fractions were obtained and combined on the basis of TLC and analyzed by HPLC. On the basis of the HPLC analysis, the composition of the mobile phase for semipreparative reverse phase HPLC was developed. The separation was achieved using a Supelco Ascentis RP-Amide (for semipreparative purposes, column length 250 × 10 mm i.d., particle size 5 µm) and Supelco Ascentis C18 (for semipreparative analyses, column length 250 × 10 mm i.d., particle size 5 µm), eluted with a gradient of MeOH, MeCN and water with 0.2% formic acid, flow rate 5 mL/min, column block temperature 40 °C (for further details of separation please see Appendix A). Fractions were collected on the basis of the response of the UV detector (λ 220, 240, 260, and 280 nm). The solvent was evaporated and pure compounds (−)-gomisin G (**2**), (−)-gomisin B (**3**), (−)-gomisin C (**4**), (−)-angeloylgomisin P (**5**), (+)-deoxyschisandrin (**6**), epigosimisin O, (−)-wuweizisu C, arisantetralone C, arisantetralone A, and (−)-schisantherin E were obtained (Appendix A) and identified (spectral data at Appendix A).

Isolated compounds (**2**, **3**, **4**, **5**, arisantetralone C, arisantetralone A, epigomisin O, and (−)-schisantherin E) were analyzed by HPLC and identified by NMR, MS, and CD. (+)-Deoxyschisandrin (**6**) and (−)-wuweizisu C were identified by comparison of UV spectra and retention times in HPLC analysis (simultaneous injection) with previously obtained lignan standards. The purity of compounds was evaluated by HPLC and exceeded 95%.

#### Isolation of Compound **1**

Fraction SC 183–205 (0.460 g) obtained from chloroform portion of *S. chinensis* methanolic extract [62] was adsorbed on 44 g of silica gel, applied on the column and separated on silica gel 60 (glass column i.d. 4.5 cm × 58 cm), which was developed with a stepped gradient using a mobile phase composed of benzene:acetone, starting composition 80:20 (*v*/*v*) to ending composition 70:30 (*v*/*v*). Fractions of 100 mL were collected. A total of 45 subfractions were obtained and combined on the basis of TLC. Subfractions VIII, IX, X–XI, XII–XIII, XIV–XV were subjected to preparative TLC using mobile phase composed of acetone:chloroform 20:80 (*v*/*v*). Additionally, 15 mg of each fraction was applied to each plate. From these subfractions, the following subfractions were obtained in the following order of retention D, D, F, and E, respectively (Appendix A). All newly obtained subfractions were evaluated by TLC (mobile phase methanol:chloroform 10:90 (*v*/*v*) and HPLC-ELSD (Appendix A). Fraction SC 183–205-X-XI-D was used for NMR analysis (Appendix A), UV (Appendix A), and HRMS (Appendix A) and proved the presence of **1**.

### 3.4. Antiproliferative Activity

The viability of THP-1™ NF-κB cells (Invivogen, San Diego, CA, USA) was measured using the cell proliferation reagent WST-1 (Roche, Basel, Switzerland) according to the manufacturer’s manual. THP-1™ NF-κB cells (floating monocytes, 500,000 cells/mL) were incubated in 100 μL of a serum-free RPMI 1640 medium and seeded into 96-well plates in triplicate at 37 °C. The antiproliferative activity was tested at a concentration of 10 µM. The compounds were dissolved in DMSO [the final concentration of DMSO in the medium was always 0.1% (*v*/*v*)]. The amount of formazan created (which correlates to the number of metabolically active cells in the culture) was calculated as a percentage of the control cells, which were treated only with DMSO and were assigned as 100%.

### 3.5. Cellular Antioxidant Activity (CAA) Assay

The antioxidant activity of the test compounds was measured in THP-1™ NF-κB cells using the method of Wolfe and Liu [34] with some modifications, as reported previously [63]. Briefly, THP-1™ NF-κB cells (floating monocytes, 600,000 cells/mL) were pre-incubated for 1 h in serum-free RPMI 1640 medium containing 25 μM 2′,7′-dichlorodihydrofluorescein-diacetate (DCFH_2_-DA; Sigma-Aldrich) dissolved in DMSO at 37 °C. After the incubation, the cells were centrifuged, washed with PBS, re-suspended in serum-free RPMI 1640 medium, and seeded into 96-well plate in triplicate (60,000 cells/well). The cells were then incubated with the test compounds dissolved in DMSO at a concentration of 5 µM for 1 h. The cells were then incubated with 600 µM 2,2′-azobis(2-methylpropionamidine) dihydrochloride (AAPH; Sigma Aldrich) to generate ROS. The plate was immediately placed into a FLUOstar Omega microplate reader (BMG Labtech GmbH, Ortenberg, Germany) at 37 °C. The level of oxidized fluorescent 2′,7′-dichlorfluorescein (DCF) was measured every 5 min for 1 h with excitation at 485 nm and emission at 538 nm. Each plate included triplicate control and blank wells: control wells contained cells treated with DCFH-DA and oxidant; blank wells contained cells treated with the dye and serum-free RPMI 1640 medium without oxidant. Quercetin was used as a positive control at the same concentration as the test compounds.

After the blank was subtracted from the fluorescence readings, the area under the curve of fluorescence versus time was integrated to calculate the CAA values of the test compounds: CAA unit = 100 − (∫SA/∫CA) × 100, where ∫SA is the integrated area under the sample fluorescence versus time curve and ∫CA is the integrated area obtained from the control curve.

### 3.6. Detection of the Activation of NF-κB

The activity of transcriptional factor NF-κB was evaluated on the THP1-Blue™ NF-κB cell line expressing on NF-κB-inducible secreted embryonic alkaline phosphatase (SEAP) reporter gene. The cells were treated with test compounds dissolved in DMSO at a concentration of 10 µM. After a 1 h incubation, cells were stimulated by LPS 0.25 μg/mL (Sigma Aldrich). The activity of NF-κB was determined after 24 h using. Quanti-Blue reagent (Invivogen, San Diego, CA, USA), as reported previously [63]. Prednisolon (Sigma Aldrich) was used as a positive control at a final concentration of 2 μM.

### 3.7. The Scrape Loading-Dye Transfer (SLDT) Assay

The Scrape Loading Dye Transfer (SLDT) assay was performed using WB-*ras* cell line. The WB-*ras* cells were developed through the transduction of WB-F344 cells with a retroviral vector containing the Ha-*ras* oncogene [64,65]. These cells were obtained from Prof. J. Trosko from Michigan State University, USA. WB-*ras* cells exhibit in vivo tumorigenicity and possess various characteristics of neoplastically transformed cells, including aberrant expression and localization of gap junction protein connexin and thus diminished levels of GJIC [64,65]. The cells were cultivated in 24-well plates (flat bottom, TPP, Trasadingen, Switzerland). Initially, cells were seeded at a density of 22,000 cells/cm^2^ and were incubated for 24 h in DMEM medium (Cat. No. 11880, Thermo Fisher) supplemented with fetal bovine serum 7.5%, *v*/*v*, Biosera, Cholet, France) and L-glutamine (2 mM, Thermo Fisher) (pH: 7.40). After the 24 h incubation period, the growth medium was exchanged with the exposure medium containing test compounds in the final concentrations of 5, 10, and 20 μM. Test compounds in stock were dissolved in DMSO, which served as the negative control (0.1%DMSO, *v*/*v*). Sodium butyrate (Sigma-Aldrich), an epigenetic modulator (histone deacetylase inhibitor) known to induce cell differentiation, inhibit cell proliferation, and induce GJIC in transformed cells such as WB-*ras* [65], was used as a positive control at a concentration of 800 μM. Cells exposed only to fresh culture medium represented the non-treated (naïve) control. Cells were exposed in triplicates for 48 h. Following this incubation, exposure medium was removed, and exposed cells were gently rinsed with calcium- and magnesium-supplemented PBS (CaMg-PBS; pH 7.2) to remove any residual medium. Subsequently, the mixture solution containing Lucifer Yellow CH dilithium salt (1 mg/mL, Sigma-Aldrich) and propidium iodide (10 μg/mL, Sigma-Aldrich) in CaMg-PBS was added to each well for cellular staining (three drops per well). The dyes were introduced to the cells by parallel cuts (three cuts per well) made by a micro knife blade, as reported previously [59]. The dye transfer was allowed for 10 min, then the staining solution was removed, and the cells were rinsed with CaMgPBS. The cells were fixed with 4% formaldehyde (*v*/*v*) in PBS.

Microphotographs of the dye transfer were obtained from each cut using Axio Observer Z1 microscope equipped with AxioCam 503 Mono camera and 10× objective. Lucifer Yellow was visualized using Filter Set 38HE (AF 488 channel: excitation—BP 450–490, beamsplitter—FT 495, emission—525/50), while propidium iodide was visualized using Filter Set 20 (Rhodamine channel: excitation—BP 546/12, beamsplitter—FT 580, emission—BP 575–640). The exposure setting was standardized across all images within a specific experiment (Lucifer Yellow: 100–500 ms; propidium iodide: 1000–2500 ms). For image acquisition, ZEN blue 3.8 software was used.

Subsequently, the images were analyzed in an open-source image processing package Fiji based on ImageJ2, using a macro which was created specifically for the evaluation of intercellular communication [59]. This macro allowed the detection of cells engaged in GJIC stained with Lucifer Yellow, along with the identification of the initially dye-loaded cells along the cut, stained with propidium iodide. The determination of GJIC was obtained by the calculations involving the subtraction of propidium iodide area (PIA) from the Lucifer Yellow area (LYA) for each cut. The net area of dye transfer for GJIC was derived from each image and compared with the corresponding values obtained for the non-treated (naïve) control from the same experiment. The result was reported as a fraction of the control (FOC):

FOC_GJIC_ = [(LYA_Treatment_ − PIA_Treatment_)/(LYA_Naïve Control_ − PIA_Naïve Control_)].


### 3.8. Statistical Analysis

Statistical analyses were carried out using IBM SPSS Statistics for Windows, software version 26.0 (Armonk, NY, USA). The data were graphed as the mean ± SEM. Comparisons between groups were made using a Mann-Whitney U test or Kruskal-Wallis test followed by pair-wise comparison with Bonferroni correction, depending on the number of experiments being compared.

## 4. Conclusions

We isolated a series of dibenzocyclooctadiene lignans and evaluated their biological activity. Furthermore, we isolated new sesquiterpene 7,7-dimethyl-11-methylidenespiro[5.5]undec-2-ene-3-carboxylic acid (**1**). Selected dibenzocyclooctadiene lignans were assayed on their anti-inflammatory potential in LPS-stimulated monocytes by monitoring their anti-NF-κB activity, antioxidant activity in CAA model, and their effect on restoring GJIC in WB-*ras* cells. The isolated compounds did not show strong activity in CAA mode. (+)-schisandrin (**7**) moderately restored the gap junction intercellular communication. The anti-inflammatory activity was proved for several schisandra lignans, with (−)-gomisin N (**8**), (+)-γ-schisandrin (**12**), rubrisandrin A (**13**), and (−)-gomisin J (**14**) showing the effects comparable with prednisolone. The results confirm the potential of the traditional medicinal plant *S. chinensis* in the treatment of inflammatory conditions.

## Figures and Tables

**Figure 1 ijms-25-03465-f001:**
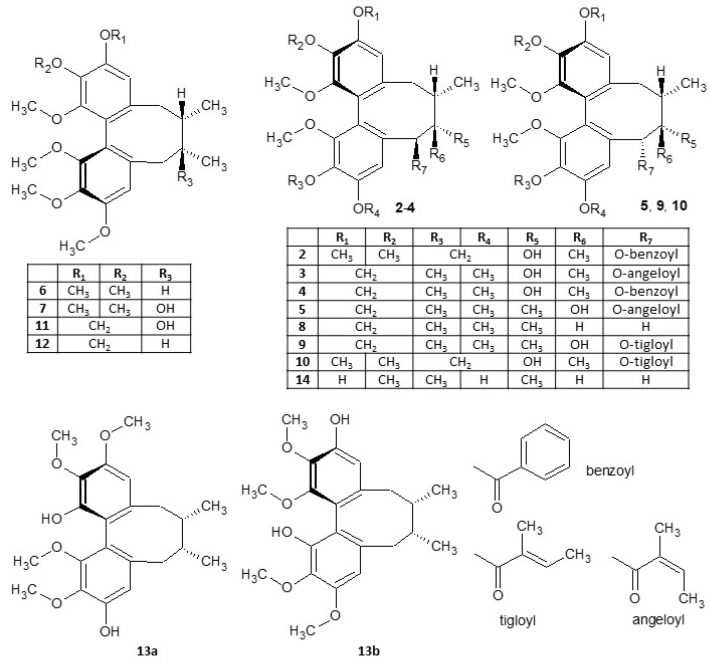
Structures of *S. chinensis* lignans.

**Figure 2 ijms-25-03465-f002:**
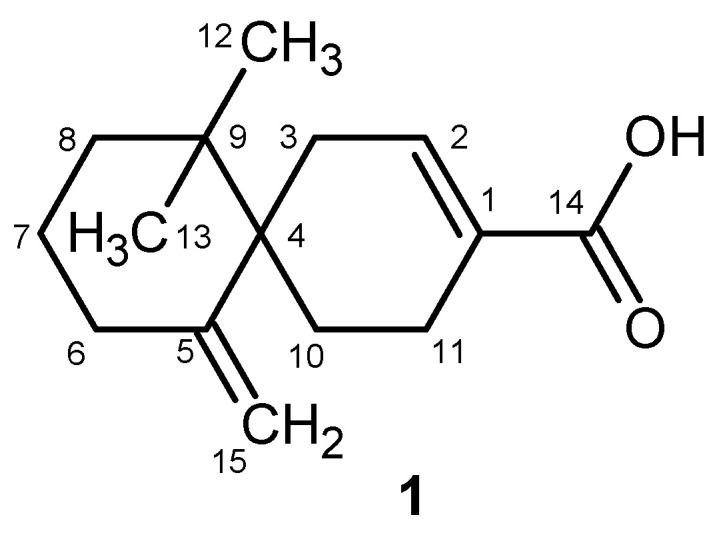
The 7,7-dimethyl-11-methylidenespiro[5.5]undec-2-ene-3-carboxylic acid (**1**).

**Figure 3 ijms-25-03465-f003:**
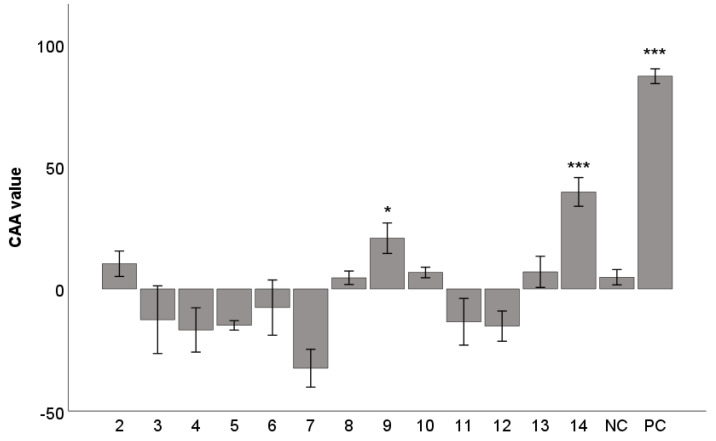
Antioxidant activity of lignans **2**–**14**, expressed as CAA values. The concentration of the test lignans **2**–**14** was 10 μM, quercetin at the same concentration was used as a positive control (PC). DMSO was used as the solvent and was added as the negative control (NC). The results are expressed as the mean ± SEM for two independent experiments measured in hexaplicate and are statistically compared to NC (* *p* < 0.05, and *** *p* < 0.001) using the Mann-Whitney U test.

**Figure 4 ijms-25-03465-f004:**
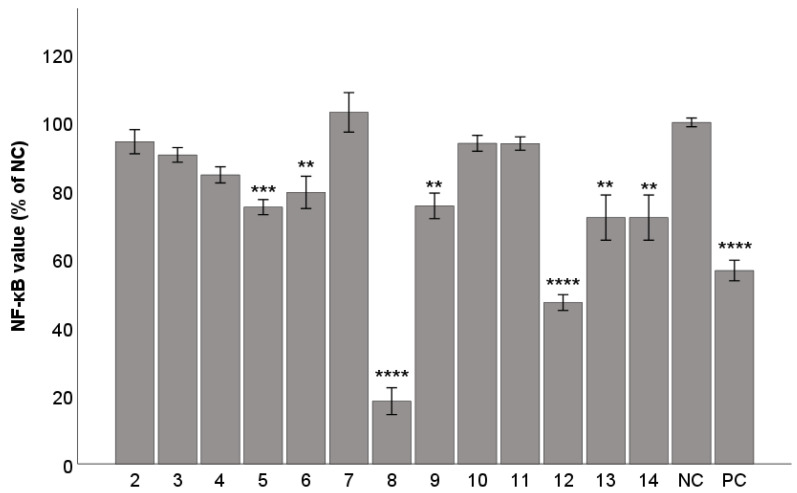
Anti-inflammatory activity of lignans **2**–**14**, expressed as inhibition of NF-κB activity. Concentration of the test lignans **2**–**14** was 10 μM, prednisolone (2 μM) was used as a positive control (PC). DMSO was used as the solvent and was added as the negative control (NC). The results are expressed as the mean ± SEM for three independent experiments measured in hexaplicate and are statistically compared to NC (** *p* < 0.01, *** *p* < 0.001, and **** *p* < 0.0001) using the Kruskal–Wallis test.

**Figure 5 ijms-25-03465-f005:**
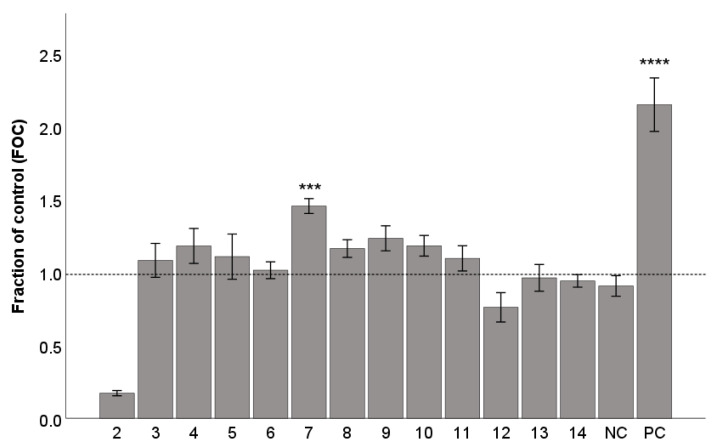
Gap junction intercellular communication (GJIC) affected by lignans **2**–**14**, expressed as a fraction of control (FOC). Concentration of the test lignans **2**–**14** was 10 μM, sodium butyrate (800 μM) was used as a positive control (PC). DMSO was used as the solvent and was added as the negative control (NC). The results are normalized to the non-treated control (FOC = 1.0), expressed as the mean ± SEM for three independent experiments measured in triplicate and statistically compared to NC (*** *p* < 0.001, and **** *p* < 0.0001) using the Mann-Whitney U test.

**Figure 6 ijms-25-03465-f006:**
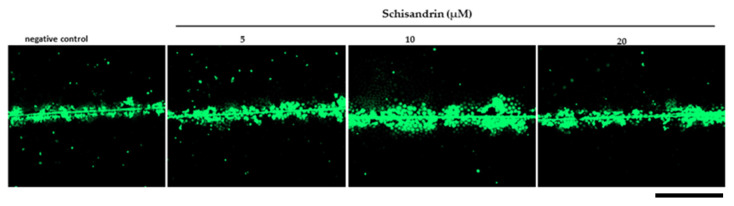
Gap junction intercellular communication (GJIC) in response to (+)-schisandrin (**7**). Representative microphotographs from the SLDT assay show the spread of the gap junction-permeable dye, Lucifer Yellow CH dilithium salt, introduced via a cut (central line) into the confluent monolayer of exposed WB-*ras* cells. DMSO was used as the solvent and served as the negative control. (+)-Schisandrin (**7**) was tested at concentrations of 5, 10, and 20 μM. Cells were exposed for 48 h. The results show that the highest induction of GJIC by (+)-schisandrin (**7**) occurred at a concentration of 10 μM. Scale bar 500 μm.

**Table 1 ijms-25-03465-t001:** The ^1^H and ^13^C NMR chemical shifts (δ_H_ and δ_C_ in ppm) of compounds **1** in CD_3_OD at 303 K.

Position	δ_H_	δ_C_
1	-	138.4
2	6.96, s	140.3
3	2.15–2.26, m	31.6
4	-	44.6
5	-	148.0
6	1.75–1.95, m	30.9
7	2.29, m	29.6
8	1.41–2.16, m	25.4
9	-	36.7
10	1.80–1.96, m	23.4
11	2.34–2.36, m	39.0
12	0.91, s	24.0
13	0.87, s	22.1
14	-	175.0
15	4.41–4.84, d, *J* = 2.2 Hz	110.0

## Data Availability

Data generated in this research are available at the authors.

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
