# Peer review of "Dibenzocyclooctadiene Lignans from Schisandra chinensis with Anti-Inflammatory Effects"

_ijms, 2024, doi:10.3390/ijms25063465_

Round 1

Reviewer 1 Report

Comments and Suggestions for Authors

1.     According to the isolation of bioactive compounds, the authors should revise the chemical structures in Figure 1, in particular cyclooctadiene and stereoconfiguration, and there is no information about compound 13 (13a and 13b) presented in this figure. Also, there is a recommendation for showing the structures of some substituents (e.g., O-angeloyl, O-tigloyl).

2.     For characterization of a new compound, all spectroscopic data should be included in the main text such as IR and MS data, yield of an isolated compound and physical appearance (if it is a solid, a melting point is required.). 

3.     According to the antioxidant activity (Figure 3), the authors should explain why the CAA values of some compounds were negative. Do these compounds cause the cytotoxicity to cells? 

4.     The concentration used in the CAA assays was at 10 µM. Did the authors conduct the cell viability based on a range of concentrations?  

5.     The authors should explain why some compounds were inactive against free radicals in the CAA assay, although some previous studies reported that those compounds had antioxidant properties.

6.     According to the anti-inflammatory activity, the authors mentioned that the anti-inflammatory activity of compounds 8 and 12 were higher than a positive control (prenidsolone). However, the concentration (10 µM) of tested compounds was different from the positive control (2 µM). The authors should clarify this point in the discussion.

7.     According to Gap junction intercellular communication, the authors should revise this sentence “The highest activity showed (+)-schisandrin (12) at a concentration of 10 μM, ……..” whether compound 12 is correct or not. Also, the explanation, why a concentration at 10 μM showed the highest activity compared to other concentrations, is required for this GJIC result. 

8.     According to the methodology (HPLC), the authors should describe the mobile phase ratio used for a gradient elution. Furthermore, chromatographic separation conditions also need to be mentioned (e.g., time, flowrate, temperature).

9.      According to the methodology (extraction and isolation), the authors need to clearly mention about the ultrasonification time, frequency and temperature performed for facilitating the extraction and the apparatus used (e.g., manufacturer). 

10.  According to the methodology (bioassays), the authors referred to other papers for the bioactivity protocols, but the procedures are needed to be clearly described in the main text. Also, the standard drug (positive control), concentration used, experiment replication, and solvent used are required for more details. The equation for calculating CAA values and FOC must be mentioned.

11. For supporting information (SI), the authors should provide peak integration of 1H-NMR spectra. There is no show of any 13C-NMR spectra. Also, the 2D NMR spectra should be presented clearly, in particular x and y axis, and peaks are not clearly seen. CD data should mention solvent and concentration used in the measurement.

Comments on the Quality of English Language

Some typos and grammatical errors should be revised. 

Author Response

Dear Reviewer 1,

we appreciate your work done towards the improvement of the manuscript. We addressed everything requested and believe that the manuscript is good enough to be published in this respected journal.

  1. According to the isolation of bioactive compounds, the authors should revise the chemical structures in Figure 1, in particular cyclooctadiene and stereoconfiguration, and there is no information about compound 13 (13a and 13b) presented in this figure. Also, there is a recommendation for showing the structures of some substituents (e.g., O-angeloyl, O-tigloyl).

Thanks for the comment. We tried to revise and improve the format of structures in Figure 1, with the aim to improve the quality of the depicting, we checked the stereochemistry at dibenzocycloocatadiene and added the structures of tigloyl and angeloyl substituents. We added the 13a and 13b structures, these were previously incorrectly removed from picture.

  1. For characterization of a new compound, all spectroscopic data should be included in the main text such as IR and MS data, yield of an isolated compound and physical appearance (if it is a solid, a melting point is required.).

Thanks for the comment. We fully understand this demand, however, we cannot add any other date in addition to analysis by 1D and 2D NMR. The amount of compound 1, which we were able to isolate prevented us from running other types of measurements. However, we believe, that the NMR data are convincing enough, as the described compound is simple and shows no stereochemistry. We added the physical appearance and amount – only 2 mg of compound 1 was obtained.

  1. According to the antioxidant activity (Figure 3), the authors should explain why the CAA values of some compounds were negative. Do these compounds cause the cytotoxicity to cells? 

Thank you for the comment. The negative CAA values in Fig. 3 are a sign of pro-oxidant effect and thus potential cytotoxic effect, however the cytotoxicity was not proved to be significant in antiproliferative assay, as you can see from supplementary data. We have added a sentence explaining this into the Discussion part.

  1. The concentration used in the CAA assays was at 10 µM. Did the authors conduct the cell viability based on a range of concentrations?

Thanks a lot for the comment. The schisandra lignans are in general taken as non-toxic. Our previous evaluations of schisandra lignans connected with their cytotoxic activity (Planta Med 2010; 76: 1672–1677; Fitoterapia 98 (2014) 241–247) did not show any strong cytotoxic effects. We tried to evaluate their cytotoxic effects at a concentration of 10 μM, and the used cell line did not show important cytotoxicity (viability observed at a minimum of 80% of control). Therefore, 10 μM was later used for CAA assay. We did not test higher concentrations, because, in general, we did not expect greater concentrations reached during administration to human or laboratory animals or during utilization of this medicinal plant. The results of the cytotoxicity assay are shown in figure S84 (Supplementary Material).

  1. The authors should explain why some compounds were inactive against free radicals in the CAA assay, although some previous studies reported that those compounds had antioxidant properties.

Thank you for the comment. The differences are caused by different methodologies used to induce oxidative stress in previously published experiments. Our method involves the testing on living cells, and we believe that it gives a better overview of activity compared to simple biochemical tests. We have added an explanation to the text.

  1. According to the anti-inflammatory activity, the authors mentioned that the anti-inflammatory activity of compounds 8 and 12 were higher than a positive control (prednisolone). However, the concentration (10 µM) of tested compounds was different from the positive control (2 µM). The authors should clarify this point in the discussion.

Thank you for the comment. We have clarified that in the manuscript.

  1. According to Gap junction intercellular communication, the authors should revise this sentence “The highest activity showed (+)-schisandrin (12) at a concentration of 10 μM, ……..” whether compound 12 is correct or not. Also, the explanation, why a concentration at 10 μM showed the highest activity compared to other concentrations, is required for this GJIC result. 

Thank you for the comment. We have changed the number and added an explanation.

  1. According to the methodology (HPLC), the authors should describe the mobile phase ratio used for a gradient elution. Furthermore, chromatographic separation conditions also need to be mentioned (e.g., time, flowrate, temperature).

Thanks for the comment, we added the details of separation into the experimental part and supporting information, we clarified chromatographic conditions for each of isolated substance.

  1. According to the methodology (extraction and isolation), the authors need to clearly mention about the ultrasonification time, frequency and temperature performed for facilitating the extraction and the apparatus used (e.g., manufacturer).

Thank you for your comment, the conditions of extraction have been added to the manuscript in experimental part (ultrasonification, 3x 1h, temperature 40 °C, frequency 35 kHz, Sonorex Digitec, Bandelin electronic GmbH & Co. KG, Germany).

  1. According to the methodology (bioassays), the authors referred to other papers for the bioactivity protocols, but the procedures are needed to be clearly described in the main text. Also, the standard drug (positive control), concentration used, experiment replication, and solvent used are required for more details. The equation for calculating CAA values and FOC must be mentioned.

Thank you for the comment. The methodology part of the manuscript was rewritten, and we have added the necessary details.

  1. For supporting information (SI), the authors should provide peak integration of 1H-NMR spectra. There is no show of any 13C-NMR spectra. Also, the 2D NMR spectra should be presented clearly, in particular x and y axis, and peaks are not clearly seen. CD data should mention solvent and concentration used in the measurement.

Thank you for your comment. We tried to add the necessary data. We did not measure 13C-NMR because these were known compounds which were described in detail earlier. Carbon shifts were obtained from HMBC and HSQC. CD data (solvent and concentration) were supplemented.

Reviewer 2 Report

Comments and Suggestions for Authors

In this manuscript, Rybnikář and colleagues assessed the anti-inflammatory and antioxidant potential of bioactive compounds in Schisandra chinensis. The focus on nutritional strategies for disease prevention and treatment is both timely and pertinent. Nevertheless, to improve clarity, specific sections necessitate additional elaboration.

Considering the special issue's focus on nutritional strategies against oxidative stress and inflammation, it's crucial to underscore, particularly in the introduction, the role of Schisandra chinensis as a dietary component.

The phrase "hundreds of studies" (lines 48 to 50) may seem somewhat ambitious given that only two works are cited. It might be beneficial to clarify that these citations are reviews, offering a broader perspective on the number of studies conducted.

The choice of a 10 μM concentration for evaluating GJIC, antioxidant, and anti-inflammatory activity needs justification. Why was this particular concentration selected, and why wasn't a dose-dependency approach considered?

The rationale behind selecting different doses (5 μM and 20 μM) for compounds (+)-schisandrin (7), (−)-gomisin N (8), and (−)-tigloylgomisin P (9) in subsequent experiments should be clarified. Also, the absence of results for other substances at these concentrations needs addressing.

Please include a graph or table in the Results section to present the Antiproliferative Activity data mentioned in the Materials and Methods section, as this information is currently missing.  

A more detailed exploration of the mechanisms, particularly how compounds 8 and 12 inhibit NF-κB, would enrich the discussion and provide a deeper understanding of their effects.

Author Response

Dear Reviewer 2,

we appreciate your work done towards the improvement of the manuscript. We addressed everything requested and believe that the manuscript is good enough to be published in this respected journal.

In this manuscript, Rybnikář and colleagues assessed the anti-inflammatory and antioxidant potential of bioactive compounds in Schisandra chinensis. The focus on nutritional strategies for disease prevention and treatment is both timely and pertinent. Nevertheless, to improve clarity, specific sections necessitate additional elaboration.

Considering the special issue's focus on nutritional strategies against oxidative stress and inflammation, it's crucial to underscore, particularly in the introduction, the role of Schisandra chinensis as a dietary component.

Thanks for the comment, we tried to put attention to dietary utilization of Schisandra fruits.

The phrase "hundreds of studies" (lines 48 to 50) may seem somewhat ambitious given that only two works are cited. It might be beneficial to clarify that these citations are reviews, offering a broader perspective on the number of studies conducted.

Thanks for the comment. We agree, and we tried to reformulate in meaning of comment.

The choice of a 10 μM concentration for evaluating GJIC, antioxidant, and anti-inflammatory activity needs justification. Why was this particular concentration selected, and why wasn't a dose-dependency approach considered?

Thank you for the comment. As described in the Methods section, we have tested at 10 µM due to the relative non-toxicity of all compounds in THP-1™ NF-κB cells (minimal viability 80% of control). We have added the figure showing the viability of cells at 10 μM concentration into the Supplementary Material.

The rationale behind selecting different doses (5 μM and 20 μM) for compounds (+)-schisandrin (7), (−)-gomisin N (8), and (−)-tigloylgomisin P (9) in subsequent experiments should be clarified. Also, the absence of results for other substances at these concentrations needs addressing.

Thank you for the comment. We have tested all compounds at a concentration of 10 µM. The four compounds showing somewhat promising results were chosen for further elucidation if lower (5 µM) or higher (20 µM) concentrations would enhance the effect, which appeared not to be the case. We tried to clarify in the text.

Please include a graph or table in the Results section to present the Antiproliferative Activity data mentioned in the Materials and Methods section, as this information is currently missing.  

Thank you for the comment. We did not normally include the data for cytotoxicity, since it is not the main aim of the article. This preliminary experiment is only to set the correct concentration for further text. We prefer to keep it in Supplementary material, and the graphic was added.

A more detailed exploration of the mechanisms, particularly how compounds 8 and 12 inhibit NF-κB, would enrich the discussion and provide a deeper understanding of their effects.

Thank you for the comment. The exact mechanism of compound 8 is not known, but we have added some details.

Reviewer 3 Report

Comments and Suggestions for Authors

The purpose of this paper is to isolate a new compound, sesquiterpene 7,7-dimethyl-11-methylidenespiro[5.5]undec-2-ene-3-carboxylic acid, and a series of dibenzocyclooctadiene lignans from Schisandra chinensis and to evaluate their biological activity. Potential biomedical applications of this work can be foreseen, and it is generally recommended for publication once the following issues listed below are carefully addressed.

1.     Structural identification is critical for the discovery of new bioactive ingredients, and some important data are missing regarding the characterizations such as MS spectroscopy, UV-vis spectrometry, and FT-IR, which are highly suggested to be supplemented. Additionally, information on yield and the purity of the isolated new compound is needed.

2.     The manuscript includes data regarding the 13 C-NMR of the new compound, but I couldn't find the 13 C-NMR Chart of compound 1 in the supplementary material, which is highly suggested to be added.

3.     In the 1H-NMR chart of compound 1, the chemical shift is detected automatically from the Mestrenova program. It is recommended to carefully check and manually detect the chemical shift of peaks related to compound 1.

4.     It was insufficient to declare the anti-inflammatory potential and antioxidant activity only for the known selected dibenzocyclooctadiene lignans. There is no data regarding the biological activity of the new compounds. Although the manuscript mentions that compounds 2-14, which are already known, were tested, the most important one and priority should be compound 1 since it is new. Therefore, it is highly recommended to test the new isolated compound and suggested to add this data.

5.     In plant materials, some data are missing and recommended to be added, such as a picture of the plant, especially during the collection process, information about who collected and identified the plant, collection time, data regarding the place of collection, and voucher number of the plant.

6.     The resolution of picture number 1 is really low, which greatly affects the readability of this manuscript. It is highly suggested that these pictures could be reconstructed.

Author Response

Dear Reviewer 3,

we appreciate your work done towards the improvement of the manuscript. We addressed everything requested and believe that the manuscript is good enough to be published in this respected journal.

The purpose of this paper is to isolate a new compound, sesquiterpene 7,7-dimethyl-11-methylidenespiro[5.5]undec-2-ene-3-carboxylic acid, and a series of dibenzocyclooctadiene lignans from Schisandra chinensis and to evaluate their biological activity. Potential biomedical applications of this work can be foreseen, and it is generally recommended for publication once the following issues listed below are carefully addressed.

  1. Structural identification is critical for the discovery of new bioactive ingredients, and some important data are missing regarding the characterizations such as MS spectroscopy, UV-vis spectrometry, and FT-IR, which are highly suggested to be supplemented. Additionally, information on yield and the purity of the isolated new compound is needed.

Thanks for the comments. We fully understand this demand, however, we cannot add any other data in addition to analysis by 1D and 2D NMR. The amount of compound 1, which we were able to isolate prevented us from running other types of measurements. However, we believe, that the NMR data are convincing enough, as the described compound is simple and shows no stereochemistry. We added the physical appearance and amount – only 2 mg of compound 1 was obtained.

  1. The manuscript includes data regarding the 13 C-NMR of the new compound, but I couldn't find the 13 C-NMR Chart of compound 1 in the supplementary material, which is highly suggested to be added.

Thanks for the comment. We commonly do not use separate 13C NMR measurement and we get the data from 2D experiments. Because we obtained only 2 mg of compound 1, we unfortunately cannot perform additional experiments. Thanks for the understanding.

  1. In the 1H-NMR chart of compound 1, the chemical shift is detected automatically from the Mestrenova program. It is recommended to carefully check and manually detect the chemical shift of peaks related to compound 1.

Thanks for the comments, we did a careful check of integration and found all relevant peaks integrated.

  1. It was insufficient to declare the anti-inflammatory potential and antioxidant activity only for the known selected dibenzocyclooctadiene lignans. There is no data regarding the biological activity of the new compounds. Although the manuscript mentions that compounds 2-14, which are already known, were tested, the most important one and priority should be compound 1 since it is new. Therefore, it is highly recommended to test the new isolated compound and suggested to add this data.

Thank you for your comment. We acknowledge that characterizing the biological activity of the newly isolated and identified compound 1, along with other lignans, would be valuable. However, the isolated quantity of compound 1 was primarily utilized for structural elucidation and wasn't sufficient for biological experiments. We believe that by publishing the new structure alongside descriptions of the anti-inflammatory, antioxidant, and GJIC-restorative activity of other Schisandra lignans, we can stimulate further interest in these compounds and encourage future studies on their biological activity.

  1. In plant materials, some data are missing and recommended to be added, such as a picture of the plant, especially during the collection process, information about who collected and identified the plant, collection time, data regarding the place of collection, and voucher number of the plant.

Thank you for your comment. it was a commercial product that was purchased from a company Salvia Paradise Ltd. (Czech Republic) in 2021. According to the company information, the fruits were produced in the northeastern Dongbei region, China, as stated in manuscript.

  1. The resolution of picture number 1 is really low, which greatly affects the readability of this manuscript. It is highly suggested that these pictures could be reconstructed.

Thanks for the comment. We tried to improve the size, added the substituents, and clarified the stereochemistry. Also, the resolution is now better. We hope that we did it well.

Round 2

Reviewer 3 Report

Comments and Suggestions for Authors

Thank you for revising the manuscript. Since you isolated the new compound, Structural identification is critical for the discovery of new bioactive ingredients, and some important data are still missing regarding the characterizations such as MS spectroscopy, 13 C-NMR Chart, UV-vis spectrometry, and FT-IR, which are highly suggested to be supplemented.

Author Response

Dear Reviewer,

thank you very much for the comment and suggestions. We fully understand and we tried to fulfil your recommendation. The most important part of the recommendation was to upgrade the support of structural elucidation of a new compound. As stated in the manuscript, we isolated only a limited amount of compound 1 (2 mg), therefore the identification process was rationalized to the basics. The NMR analysis was carried out with the utilization of 2D experiments, which allowed us also to understand chemical shifts in the carbon spectrum. This is a common approach, which we utilize generally for our measurements, and which decreases the time and mass demand on our compounds. Unfortunately, we do not have the amount which would allow us to add the classical one-dimensional carbon spectrum. However, from the evaluation of minimal material residuum after isolation and identification, we had been able to process HR-MS analysis using a described method on orbitrap, and this method, spectrum, and values are now added to the manuscript and supplementary file to support the structural elucidation of compound 1. Furthermore, we tried to extract the UV spectrum of compound 1, however, as you can see from the supplementary material, it is non-specific and not helpful for the identification.

Dear reviewer, we hope that this improves the manuscript enough to be now accepted for publication.

Round 3

Reviewer 3 Report

Comments and Suggestions for Authors

Thank you for revising the manuscript. In my opinion, it's suitable for publication. Thanks!